# Managing Oral Health in the Context of Antimicrobial Resistance

**DOI:** 10.3390/ijerph192416448

**Published:** 2022-12-08

**Authors:** Lucinda J. Bessa, João Botelho, Vanessa Machado, Ricardo Alves, José João Mendes

**Affiliations:** 1Centro de Investigação Interdisciplinar Egas Moniz (CiiEM), Egas Moniz—Cooperativa de Ensino Superior, Caparica, 2829-511 Almada, Portugal; 2Clinical Research Unit (CRU), CiiEM, Egas Moniz—Cooperativa de Ensino Superior, Caparica, 2829-511 Almada, Portugal; 3Evidence-Based Hub, CiiEM, Egas Moniz—Cooperativa de Ensino Superior, Caparica, 2829-511 Almada, Portugal

**Keywords:** antibiotic resistance, antibiotic stewardship, dental practice, oral infections, oral microbiome, oral resistome

## Abstract

The oral microbiome plays a major role in shaping oral health/disease state; thus, a main challenge for dental practitioners is to preserve or restore a balanced oral microbiome. Nonetheless, when pathogenic microorganisms install in the oral cavity and are incorporated into the oral biofilm, oral infections, such as gingivitis, dental caries, periodontitis, and peri-implantitis, can arise. Several prophylactic and treatment approaches are available nowadays, but most of them have been antibiotic-based. Given the actual context of antimicrobial resistance (AMR), antibiotic stewardship in dentistry would be a beneficial approach to optimize and avoid inappropriate or even unnecessary antibiotic use, representing a step towards precision medicine. Furthermore, the development of new effective treatment options to replace the need for antibiotics is being pursued, including the application of photodynamic therapy and the use of probiotics. In this review, we highlight the advances undergoing towards a better understanding of the oral microbiome and oral resistome. We also provide an updated overview of how dentists are adapting to better manage the treatment of oral infections given the problem of AMR.

## 1. Introduction

Oral health is a paramount factor influencing not only one’s general health but also well-being and quality of life [1]. Paradoxically, oral health management deals with a variety of diseases and conditions, ranging from dental caries and periodontal disease to tooth loss and oral cancer.

Over the years, researchers have used different definitions when referring to the microbiome [2]. However, due to the emergence of next-generation sequencing technologies and bioinformatics tools, it is possible to refine the definition of the microbiome. Considering the current knowledge, the microbiome may be defined as a dynamic and interactive micro-ecosystem predisposed to change in time and scale that is integrated into a macro-ecosystem, the host [2]. Within the human microbiome, the oral microbiome is one of the most diverse and plays a major role in defining oral health state. When the oral microbiome is perturbed, dysbiosis is installed, with disease-promoting bacteria taking over and causing an oral disease condition [3]. Therefore, the primordial goal for dental practitioners and patients alike is to preserve or restore a balanced oral microbiome. Moreover, the role of the oral microbiome in a range of non-oral diseases, including pancreatic cancer and diabetes mellitus, has been revealed [4,5,6].

The microorganisms composing the oral microbiome tend to be organized in biofilms, which are structurally and functionally complex polymicrobial communities embedded in an extracellular matrix and adhered to the hard tissue of teeth (but also on restorative materials, dental prostheses, and dental implants) and on the soft tissue of the oral mucosa [7]. Oral pathogenic biofilms have been recognized as a predisposing factor for several oral infections, including gingivitis, dental caries, periodontitis, and peri-implantitis [8,9].

Within biofilms, bacteria are more protected from the action of the host immune system, environmental factors, such as shear stress, and antimicrobial agents [10,11]. Due to the increased tolerance to antibiotics and the underlying polymicrobial nature of oral biofilms, the treatment of biofilm-related oral diseases is rather challenging [9]. Because of this, antibiotic combination therapies have been mostly applied and complemented with mechanical debridement, scaling, and root planing (SRP) [12]. Nonetheless, the success of such treatments is not always achieved, resulting in the recurrence of the infection.

Accounting for the difficulty of effectively treating oral biofilm-related infections with antibiotics is the problem of antimicrobial resistance (AMR). According to the FDI World Dental Federation, about 10% of all antibiotics for humans are prescribed by dentists [13]. Thus, optimizing the use of antibiotics through antibiotic stewardship in dentistry is highly recommended. The development of antibiotic prescribing guidelines for dentists has been the main intervention to achieve that purpose. Additionally, advancing new effective antibacterial/antibiofilm solutions and strategies are required and are being currently pursued [8].

This review aims to provide an updated overview of antibiotic use and the effect of new alternative solution therapies on the management of oral infections given the problem of AMR.

## 2. Methods

The studies included in this review were retrieved from various databases, but mostly from PubMed and Scopus. A literature search was performed using various keywords, among which we highlight the following: oral microbiome, antibiotic resistance, resistome, oral infections, dental plaque, periodontitis, peri-implantitis, and alternatives to antibiotics. No restrictions were applied on article type and there were no inclusion criteria. Rather, the appropriateness of each identified article was determined by the authors who aimed to include a wide and unbiased range of relevant and recent studies. Most studies included were published from 2015 onwards.

## 3. Oral Microbiome—A Quick Overview

The term microbiome has been connected to various definitions, but it has been recognized that there is a need to converge to a clear and commonly agreed upon definition of the microbiome [2]. Considering the recent definition proposed for microbiome by Berg et al. [2], we may define the oral microbiome as a dynamic ecosystem composed of the oral microbiota, viruses, their structural elements, and metabolites, as well as molecules produced by the coexisting host and controlled by the surrounding environment. The oral microbiome, like every microbiome, is continuously and functionally evolving due to the microbe–host and inter-species interactions.

In the oral cavity, there are distinctive niches, each with its associated microbiome, including the gingival sulcus, tongue, cheek, hard and soft palate, floor of the mouth, throat, saliva, teeth, and, if present, dental implants [3,14,15,16]. The Human Microbiome Project defined nine anatomical locations in the human mouth in a state of health: the tongue dorsum, the hard palate, the tonsils, sub- and supra-gingival plaque on teeth, the keratinized gingiva, the buccal mucosa, the throat, and saliva [17].

The microbial oral community shifts constantly throughout an individual’s life due to extrinsic and intrinsic factors [3,18,19,20,21]. The oral cavity is exposed to exogenous microorganisms mainly through diet, drinking, air, kissing, and lifestyle [18]. Changes in oral pH, depressed immune system, presence of chronic diseases, and intake of antibiotics also affect the composition of the oral microbiome [19,21]. Nonetheless, despite all these factors accounting for inter-individual variability, an oral core microbiome has been pinpointed [14,22], and it refers to the microbial taxa or the genomic and functional attributes associated with those taxa that are characteristic of the oral cavity under healthy conditions [23,24].

The human oral core microbiome has been typically identified and quantified by the occurrence and/or relative abundance of core microbial taxa across multiple samples of the same host/environment using the 16S rRNA gene sequencing method [23]. Based on that method, the expanded Human Oral Microbiome Database [25] assembled 774 oral bacterial species, of which 58% are officially named; *Streptococcus* spp. is the most prevalent genus, but species of the genera *Veillonella* spp., *Prevotella* spp., and *Fusobacterium* spp. are also common [5]. That approach is, however, somehow narrow since it unveils only a taxonomic rather than a functional core microbiome. Next-generation sequencing methodologies have advanced in recent years. Currently, shotgun metagenomics and metatranscriptomics are useful tools to provide in-depth insight into the functional profiling of microbiomes, both in health and disease states [26,27]. It is noteworthy that these advanced sequencing methodologies incur a higher cost of equipment and resources, and that may constitute a limitation to their broad application. It is believed that such technologies will tend to become more accessible, especially after the changes experienced in real time over the course of the COVID-19 pandemic [28].

Quantitative and qualitative variations in the composition of the core microbiome cause dysbiosis, which correlates with the disease state [29,30]. The taxonomic composition of microbial communities implicated in dental caries, periodontitis, and peri-implantitis has been pinpointed by several studies throughout the years [30,31,32,33,34,35] (Figure 1), which have disclosed that the microbial communities are distinct in those three oral diseases in terms of composition and/or abundance. Nonetheless, the taxonomic composition is unsurprisingly not definite; it is being constantly updated as knowledge and technologies are advancing. 

Despite the interpatient variability in microbial composition between healthy individuals and diseased patients, disease-associated microbial communities display conserved metabolic profiles that are generally accomplished by a patient-specific microbiome [36]. A recent study seems to support a cooperative metabolic interaction between common constituents of the oral microbiota in periodontitis. In detail, *Fusobacterium nucleatum* seems to use amino acids released by *Streptococcus gordonii* to produce polyamines that foster the overgrowth and spread of *Porphyromonas gingivalis*, a keystone periodontal pathogen [37]. Hitherto, these new recent breakthroughs in oral microbiome still require a clinical translation towards precision dentistry, which is still in its early stages [38]. This approach uses patients’ risk factor data beyond the clinical presentation for personalized prevention, early diagnosis, and predictive treatment [30,38].

## 4. Oral Biofilms and Biofilm-Related Oral Diseases

Bacteria play a major role in biofilm formation as the initial colonizers and in terms of abundance and function in a normal oral microbiome. However, fungi, viruses, archaea, and protozoa are also constituents of the oral community and must be taken into consideration when studying the complete oral microbiome [5]. *Entamoeba gingivalis* and *Trichomonas tenax* are the most commonly found protozoa in the oral cavity, while *Candida* species are the most prevalent fungi [18].

Bacteria within a biofilm communicate via quorum sensing, through which signaling molecules are produced and detected by neighboring bacteria. This communication system enables bacteria to regulate several bacterial mechanisms, such as the production of virulence factors and biofilm formation [39]. In addition, biofilm seems to boost bacterial protection against the host immune system, environmental factors (such as shear stress), and antimicrobial agents [10,11]. Thus, any biofilm-associated infection, including those of the oral cavity, represents a therapeutic conundrum.

*Streptococci* and some *Actinomyces* species are known to be the early colonizers of the salivary pellicle, an organic film composed mostly of proteins on the tooth surface, epithelium, and restorations [32,40]. The colonization of the salivary pellicle is the starting point for the co-aggregation of new species to previously adhered bacteria and the subsequent formation of a polymicrobial biofilm known as dental plaque, which is a natural phenomenon involved in the physiology and defenses of the host [41], as long as a certain degree of stability, known as microbial homeostasis, is maintained despite regular environmental perturbations, such as dietary intake and oral hygiene. Consequently, dysbiotic dental plaque is implicated in the development of common oral diseases, such as erosion, dental caries, periodontal disease, and peri-implantitis [40], and is characterized by an imbalance in the biofilm composition favoring oral pathogens to take the lead. It is noteworthy that the appearance and persistence of dysbiosis rely on both microbial changes and host factors, namely the development of inflammation and the intake of dietary sugars [42].

Dental caries is a multifactorial oral disease, which is biofilm-mediated and modulated by dietary carbohydrates [43]. High and frequent exposure to fermentable carbohydrates is the driver for the development of a supragingival dysbiotic biofilm, where aciduric bacteria prevail and lead to a pH decline that can no longer be buffered by saliva. Therefore, there is a selection for more acid-tolerant microorganisms, which in turn favors the persistence of this acidic environment that enables enamel demineralization [42,44,45]. In early studies, *Streptococcus mutans* has been recognized as ‘the cariogenic keystone pathogen’; however, next-generation sequencing (NGS) and omics studies have disclosed that dental caries is more a polymicrobial disease based on a variable and diverse pathogenic community that relies on sugar consumption, rather than a classic Koch’s postulate (a single agent related disease) [45,46]. Despite the substantial inter-individual variability composition of this dysbiotic biofilm in dental caries, different microbial combinations have been identified to have a similar functional profile [47]. This suggests that the pathogenesis of dental caries is controlled by complex and intricate host, microbial, and environmental factors and interactions, and for that reason, it remains an entangled research issue [45].

Periodontitis and peri-implantitis are also biofilm-mediated oral diseases with marked microbial dysbiosis and inflammation. In both, alveolar bone is lost, in addition to the loss of tooth-supporting tissue in periodontitis and peri-implant tissue in peri-implantitis, respectively [48]. If untreated, these oral non-communicable diseases can lead to tooth or dental implant loss. On the one hand, both diseases share many etiological and clinical features. On the other hand, they differ in the microbial community present. This is mostly due to the material (dentin or titanium) that serves as the substratum for biofilm formation and affects bacterial adhesion, determining differences in the type of initial bacterial colonizers and dictating the formation of distinct subgingival biofilms [49]. 

A recent study by Komatsu et al. [35] demonstrated differences in microbial and gene transcription activity in peri-implantitis and periodontitis by using integrated metagenomic, metatranscriptomic, and network data analysis. Thus, it is now evident that the microbiome in peri-implantitis is distinct from that in periodontitis. Moreover, Dabdoub et al. [50] demonstrated, through the use of 16S rRNA gene sequencing, that adjoining peri-implant and periodontal biofilms in states of health and disease present a significant dissimilarity. A more recent study [48] used metagenomic sequencing and also confirmed that the community structure in healthy and diseased microbial communities around teeth and implants is distinct. Another similar study [51] reached the same conclusions.

With the progress of metagenomics and metatranscriptomics studies on oral health and disease states, the composition and structure of microbial communities, as well as the overall functional characteristics of the microbiome, can be profiled and will pave the way to providing biomarkers that, in combination with clinical and biological parameters, will be useful for disease diagnosis and management [52].

## 5. Antimicrobial Resistance (AMR) in Dental Practice

AMR is one of the most relevant problems of Public Health nowadays since it endangers prophylaxis and infection treatment. Among the antimicrobial agents, antibiotics are by far the most prescribed. The extensive use (and most often abuse and misuse) of antibiotics in the sectors of human and veterinary medicines, agriculture, aquaculture, and livestock farming has accelerated the emergence and spread of antibiotic resistance mechanisms by bacteria [53]. Recently, awareness of this problem has been publicized and specific actions are being demanded to be taken. It is consensual that AMR needs to be tackled by a ‘One Health’ approach with integrated actions across human, animal, and environmental health sectors [54].

A rather high percentage of around 10% of all antibiotics for humans is prescribed by dentists, according to the FDI World Dental Federation [55]. Therefore, dentists have a vital role to play in lessening antibiotic prescribing and using antibiotics more appropriately [56].

### Antibiotic Stewardship in Dentistry

Dentists prescribe antibiotics for two purposes: (1) prophylaxis, to improve the outcome success of surgical interventions and reduce complications and symptoms, and (2) therapeutics, for treating oral infections [57]. However, in dental practice, antibiotic indication has been long based on personal experience or judgment and on old evidence, rather than on effective diagnosis [58]; and it has been mostly an empirical drug prescription, with a predominant choice for broad-spectrum antibiotics [57,59]. Furthermore, guidelines for the prudent usage of antibiotics were scarce or not generally shared among dental practitioners before [60]. Due to the advent of AMR, guidelines and recommendations have been created and updated by multiple national and international dental associations and expert groups [57,61,62,63,64,65,66]. Nonetheless, the guidelines for prudent antibiotic use can vary between countries, recommending different approaches based on the antibiotic resistance patterns known nationwide. In addition, therapeutic and prophylactic indications for dental antibiotics are not consensual and differ internationally [67].

Therefore, it is evident that the implementation of antibiotic stewardship programs in the dental setting is of great need [68]. Antibiotic stewardship entails a set of coordinated interventions to promote the correct use of antibiotics (optimal selection, dosing, route, and duration of administration), to improve clinical outcomes and minimize side effects to patients, and to reduce the development and spread of multidrug-resistant bacteria [69].

A multidisciplinary antibiotic stewardship team, involving the dental team in close collaboration with pharmacists, microbiologists, or other health care professionals, is fundamental to assure the execution of an antibiotic stewardship program in a dental setting [70]. A set of antibiotic stewardship interventions recommended for dental practice are compiled in Table 1.

Among these interventions, the education of both dentists and patients on the adequate use of antibiotics and the current problem of AMR should be encouraged. Several studies have confirmed the critical need for dental students’ education and specific training in the prescription of antibiotics [71,72,73,74]. Educational interventions can include lectures, didactic meetings, workshops, and practice campaigns. A recent study showed that final-year undergraduates from the Glasgow Dental School were enthusiastic about attending a supplemental, yet mandatory, online course on the essential role of dental teams in antimicrobial stewardship and in reducing AMR; the students’ feedback was positive, and they recognized they could play an important role in stewardship [75]. Evidence from another study also attested to the success of educational programs for dental students about appropriate antibiotic prescribing [76]. Education, an apparent basic intervention, is essential not only in undergraduate courses but also in postgraduation dentistry courses, and there must be an ongoing learning process for all established dentists as well.

The need for an international consensus on a core outcome set for dental antibiotic stewardship has been recently acknowledged, and its concretization will be particularly important in achieving a uniform way to ease the comparisons of dental antibiotic prescribing between studies and countries [67].

It is noteworthy that the application of antibiotic stewardship interventions in dental practice also matches the idea of precision medicine in dentistry since it is one of the steps of personalized therapy aiming at improving efficacy and reducing adverse effects [57].

## 6. Antibiotic Prophylaxis (AP) and Treatment of Oral Infections

### 6.1. Recent Changes for AP

As we have noted, the choice to use antibiotics must be based on a thorough medical history, physical examination, and diagnosis. In dental practice, antibiotics can be indicated for prophylaxis, either primary (to prevent surgical onsite infections) or secondary (to prevent infection at a distant site) [59].

The actual tendency, mostly to lessen the development of AMR, is to reduce the use of prophylactic antibiotics. Current evidence has led to the recommendations [61] to not use antibiotics to prevent postoperative complications after (1) extraction of impacted wisdom teeth [77], (2) surgical extractions of teeth or retained roots, (3) minor surgical removal of soft tissue lesions, and (4) peri-radicular surgery. Moreover, antibiotics are not recommended for the prevention of pain associated with irreversible pulpitis [64].

Undergoing AP prior to dental procedures has been a common procedure and is currently indicated by American and European guidelines on the recommended use of antibiotic prophylaxis to prevent infective endocarditis (IE) in subjects undergoing invasive procedures [65]. Nowadays, Sweden and the United Kingdom have, however, abandoned the use of AP in dentistry for the prevention of IE [78]. A nationwide cohort study conducted in Sweden provided results suggesting that the Swedish recommendation of 2012 to not administer antibiotics in dentistry for the prevention of IE did not cause an increased incidence of oral streptococcal IE among high-risk individuals [78]. Nonetheless, another recent study by Rutherford et al. [79] did not lead to clear evidence on whether AP was effective or ineffective against bacterial endocarditis in at-risk patients who underwent dental procedures. Yet, a recent cohort study on almost eight million American subjects found a significant temporal association between dental procedures considered invasive (mainly, extractions and oral surgical procedures) and IE in individuals at high risk [80]. Furthermore, this study from the University of Sheffield also found a significant decrease in IE events when AP was delivered in such invasive treatments [80]. With this discrepant evidence, the American Heart Association guideline on AP is under discussion, and, for this reason, more studies and expert discussions are needed to arrive at an adequate and definite recommendation.

Other recommendations have also been made for AP, for instance, for prosthetic joint patients undergoing invasive dental procedures. Nevertheless, it is becoming evident that there is no reason to administer antibiotics before dental procedures in patients with prosthetic joints [81,82,83].

### 6.2. Antibiotic Treatment of Dental Caries, Periodontal Diseases, and Peri-Implantitis

The administration of antibiotics in dental care is a common procedure. However, due to the increase in antibiotic resistance and recent advances in scientific knowledge, there has been a transformation in how dentists clinically manage the most common conditions, particularly dental caries, periodontal diseases, and peri-implant conditions.

In 1946, McClure and Hewitt first reported penicillin to inhibit caries in rats [84], and, in 1950, similar results were found in children [85]. Ever since, there was an exponential increase of penicillin as the first choice not only in dental caries but also in other conditions, such as periodontal diseases [86]. However, the certainty of their clinical effectiveness in terms of costs and benefits has never been clarified. Concerning dental caries, systemic antibiotics fell into disuse when the focus shifted to the antimicrobial properties of dental materials used to restore teeth (both adhesive and restorative ones) [87]. In addition, there is insufficient evidence to recommend a particular material based on its antibacterial effects in cavity lining procedures [88].

Regarding periodontal diseases, the use of systemic antibiotics become popular in aggressive forms of periodontitis or ulcerative periodontal conditions based on the aggressive clinical progression of these diseases. On the one hand, systemic antibiotics do impact periodontal parameters in the short-term without a significant change in serum markers [89,90], and with higher side effects [90]. However, for long-term follow-up, there is low-certainty evidence that systemic antibiotics could be of help in the treatment of periodontitis [91]. Moreover, there is very low-quality evidence on the effects of systemic antibiotics in adults with symptomatic apical periodontitis or acute apical abscess [92]. Considering the low evidence on several dental procedures and the potential consequences (side effects and AMR), alternatives have been proposed, such as antimicrobial peptides in dental caries, with some showing remineralizing properties with hydroxyapatite [93], and in periodontitis [94,95].

There is no specific recommendation for peri-implantitis treatment; therefore, several surgical and nonsurgical treatment approaches have been used [96]. Even if the non-surgical procedures (i.e., mechanical debridement) have been largely used to treat peri-implantitis, it has soon become evident that such an approach is not sufficient by itself to completely remove the biofilm from the implant surface. Therefore, additional/adjunctive treatment approaches have been used, such as topic or systemic antibiotics, photodynamic therapy, or surgical therapy [96,97]. Despite the use of topical/systemic antibiotics as an adjunctive therapy in peri-implantitis, there is a lack of consensus and evidence regarding their real efficacy in the definitive treatment of the disease [98], and thus, their use has not been advised for that purpose.

## 7. Oral Resistome

The emergence and spread of antibiotic resistance inevitably affect the bacteria of the human oral microbiome, making it a reservoir of antibiotic resistance [27,99,100].

The usage of antibiotics is undoubtedly an important driver for the development of antibiotic resistance genes (ARGs). The presence of oral pathogenic bacteria harboring ARGs endangers the success rate of antibiotic treatment recommended for certain oral infections, while commensal bacteria carrying ARGs may be responsible for antibiotic infections at other sites of the human body [27,101]. Based on that, a broader concept arose which is oral resistome. The term resistome dates back to the year 2006 [102], and it is nowadays defined as the collection of all the ARGs and their precursors in pathogenic and nonpathogenic bacteria composing a microbiome [103].

As such, examining the composition and changes of the oral resistome holds a promising interest in order to uncover oral antibiotic resistance profiles, circumvent failure of antibiotic treatment, and develop new effective therapies [101]. To that end, high-throughput next-generation sequencing technologies, such as metagenomics, are paving the way to an in-depth understanding of the distribution and diversity of ARGs in the oral microbiome, in addition to the disclosure of novel ARGs [26]. In particular, shotgun metagenomics is deemed essential to map all the resistance genes in the oral microbiome as well as to predict the function of these genes [104]. To date, however, there still is little understanding of the oral resistome, in both health and disease states, with only a few studies providing some insights so far, as will be described next.

Recently, the study by Caselli et al. [31] quantified ARGs in oral rinse specimens from 20 individuals with good oral health (no oral disease) using real-time quantitative PCR microarray analysis. A high prevalence of genes conferring resistance to macrolide, lincosamides, streptogramin, and tetracycline was found. Of note, DNA microarrays are limited by low throughput, usually targeting known ARGs.

Another study [101] used shotgun metagenomics to derive 788 and 386 resistomes from the oral and gut samples, respectively, obtained from healthy individuals from several countries worldwide. They found that the prevalence of ARGs was different between countries and between the oral and stool samples; oral resistome contained a higher abundance and lower diversity of ARGs, compared to the gut resistome.

Kang et al. [99] investigated for the first time the profile of the microbial community and antibiotic and metal resistance genes (MRGs) in dental plaque from metagenomic sequences deposited in a database, which were previously obtained from periodontal health and disease samples and after SRP therapy. They noticed that periodontitis and SRP treatment lad to a higher abundance of ARGs and MRGs in dental plaque and significantly altered the composition of ARG and MRG profiles. Moreover, the ARGS conferring resistance to bacitracin, beta-lactams, macrolide-lincosamide-streptogramin (MLS), and tetracycline were the most prevalent, whereas the most abundant MRGs were those conferring resistance to multi-metals, iron, chromium, and copper. Importantly, based on their results, they could assume, before the co-occurrence of ARGs, MRGs, and mobile genetic elements (MGEs), that a co-selection phenomenon may exist in the resistome of dental plaque.

Since the oral microbiome is a reservoir of antibiotic resistance, it favors the horizontal gene transfer of ARGs and MGEs. Thus, monitoring the oral resistome has huge potential in providing the reference levels for proper antibiotic use and helping the development of new and more effective antimicrobial strategies for the treatment of oral infections, namely periodontitis and peri-implantitis.

## 8. Current Alternatives to Antibiotics to Prevent and Treat Oral Infections

As most infections in the oral cavity are biofilm-associated, one obvious strategy entails the prevention of biofilm formation by improving oral hygiene practices with fluoride toothpaste [105]. However, this approach is not sufficient per se to alleviate the burden of oral biofilm-related infections. Therefore, dental materials (for dental restoration procedures or dental implants) with antimicrobial properties have been developed over the past few years. According to Jiao and collaborators [8], antimicrobial dental materials have been advanced based on three mechanisms: antimicrobial agent release, contact killing, and multi-functional strategy. Materials can incorporate antibiotic drugs or other antimicrobial agents, such as antimicrobial peptides and silver nanoparticles [8]. It is opportune to highlight, at this point, the impact of nanoengineering strategies on the significant increase of new or improved dental nanomaterials with antibiofilm properties [106,107]. In fact, nanotechnology allows the development of antimicrobial dental nanomaterials, such as those containing silver, zinc, or copper nanoparticles [108]. Though in vitro research has revealed the promising efficacy of these innovative antimicrobial materials, a major challenge with regard to the biocompatibility of these materials has been limiting their clinical application. A thorough overview of the evolving trends and developments in antimicrobial materials for dental restorative, reconstructive, and replacement strategies has been recently undertaken by Ramburrun et al. [109].

Alternative strategies to antibiotics for controlling oral infections exist and have been investigated; most of them have already been tested clinically (Figure 2). Some of those strategies include the use of antimicrobial photodynamic therapy (APDT), cold atmospheric plasma (CAP), probiotics, natural products, inhibitors of virulence factors, antimicrobial peptides, and others. Combinations of these strategies have also been explored [110].

### 8.1. Antimicrobial Photodynamic Therapy (APDT)

APDT is an emerging and non-invasive treatment method, involving a photosensitizer and a low-energy laser light in the presence of oxygen to generate reactive oxygen species (ROS) that are responsible for the bactericidal effect [8]. The potential use of APDT in dentistry has recently been explored [111], with several in vitro and clinical studies being conducted to support the therapeutic application of APDT alone, but mostly as an adjunctive treatment, in dental caries [112,113], endodontic diseases [114,115], periodontal diseases [116,117], and peri-implantitis [118,119]. There are several synthetic molecules that can be used as photosensitizers (toluidine blue, methylene blue, and indocyanine green) and combined with lasers of various wavelengths. Natural photosensitizers, such as curcumin, chlorella, chlorophyllin, and phycocyanin, among others [120], are less toxic and costly than synthetic ones and have also started to be used in APDT against oral pathogenic bacteria, with encouraging in vitro results obtained in many recent studies that stimulate research in this area to be continued and advanced. Moreover, the nanoparticles of these photosensitizers are being developed either to enhance their photosensitivity or to increase their solubility [121,122].

APDT holds great potential and utility in the field of dentistry; nonetheless, further clinical studies need to be conducted before it can be approved for therapeutic dental applications [8,111].

### 8.2. Cold Atmospheric Plasma (CAP)

CAP is another innovative strategy showing significant advantages over traditional antibiotic approaches and with good prospects to be adopted for clinical dental applications to control biofilm infections [8]. CAP generates reactive oxygen and nitrogen species that diffuse into the biofilm and cause oxidative damage to the bacterial membrane, and to the extracellular DNA and proteins constituting the biofilm [110]. There is also some evidence pointing towards an effect of CAP on inhibiting the bacterial quorum sensing, and, thus, attenuating the virulence factors of pathogenic bacteria, such has *Pseudomonas aeruginosa*; however, the exact mechanism of action is yet to be clarified [123].

### 8.3. Natural Products

Nature has been a prolific source of antimicrobial compounds, namely antimicrobial peptides. A significant number of natural product drugs/leads are produced by microorganisms and/or microbial interactions; thus, the field of natural product research must continue to be explored as a source of compounds with diverse desirable bioactivities [124].

Many natural products have been attributed the capacity to interfere with quorum sensing (QS), and, therefore, they can be useful to treat microbial infections since they can hamper biofilm formation and diminish the expression of virulence factors. Plant-derived natural products, which include phenolics, quinones, flavonoids, alkaloids, and terpenoids, have been deemed as new sources of quorum sensing inhibitors (QSI) [125]. The treatment of oral biofilm-related infections may also benefit from these QSI. For instance, a recent study showed that coumarin could inhibit *P. gingivalis* biofilm formation through a QS system [126]. In another study, curcumin decreased the expression of biofilm formation and virulence-related genes in both mono- and dual-species biofilms of *S. mutans* and *C. albicans* [127].

Chauhan et al. [128] provided an insightful overview and examples of the current use of natural products in dentistry, highlighting their major utility as adjunctive therapeutics. As we have already mentioned before for other therapies, natural products can also be enhanced by nanotechnology tools, combined with antibiotics, or combined with APDT [129]. The major drawback regarding natural products, especially herbal products, has been their cytotoxicity, which is usually due to the high dose that is required for the therapeutic action [34,129].

### 8.4. Antimicrobial Peptides (AMPs)

Another promising alternative to antibiotics in the treatment of microbial infections, including those of the oral cavity, is AMPs. AMPs are oligopeptides with a varying number (usually less than a hundred amino acids) that can be produced by most living organisms, ranging from bacteria to humans, as the first line of defense, but they can also be chemically synthesized [130]. In addition to their broad-spectrum activity against many microbes (bacteria, fungi, viruses, and parasites), other activities, such as antioxidant and antitumor activities, have been attributed to these peptides. AMPs can present more than one mechanism of action simultaneously, leading to the direct killing of microorganisms, while also modulating the immune response [131]. Having multiple mechanisms of action makes the development of resistance more difficult [132].

There are also AMPs with the ability to specifically affect biofilms by inhibiting biofilm formation or destroying preformed biofilms by interfering with the synthesis of the extracellular polymeric substance (EPS) matrix or with quorum sensing [133].

There are still some challenges for AMPs to be applied in the medical field, namely the lack of selectivity against specific strains and their susceptibility to proteases and extreme pH (e.g., gastric pH). In that regard, nanoparticles can be exploited to enhance their proteolytic stability and improve their activity and biocompatibility [133].

Combinations of AMPs and other antimicrobial agents are also being explored to better target multispecies biofilms. For instance, the new peptide DJK-5 in combination with EDTA showed a stronger bactericidal effect than alone against endodontic multispecies biofilms [134].

### 8.5. Probiotics

In recent years, a significant number of studies and systematic reviews have been suggesting and showing the benefits of using probiotics as a preventive and therapeutic method for oral infections [135,136]. Several mechanisms of action have been assigned to probiotics to account for their potential as therapeutic treatments for oral diseases, including hampering pathogens colonization by direct competition for nutrients or binding sites, producing antibacterial compounds (e.g., bacteriocins) targeting specific oral pathogens, altering oral biofilm composition and function, modulating signaling and metabolic pathways locally, and improving the immune-inflammatory response [135,137].

A growing amount of evidence has been showing that *Lactobacillus* and *Bifidobacterium* strains can alter subgingival biofilms and modulate the immunoinflammatory response, leading to improved periodontal health [138,139,140]. Very recently, Wu and collaborators [140] showed, through metagenomic sequencing, that *Lactobacillus paracasei* L9 and *Bifidobacterium animalis* A6 proportionate a significant modification on the taxonomic composition of the subgingival microbiome, by reducing the relative proportions of pathogenic bacterial genera, such as *Streptococcus*, *Fusobacterium*, *Veillonella*, and *Escherichia*; overall, they recommended additional research to be conducted to explore the promising use of these two probiotics as prophylactic agents for periodontitis.

Though probiotics are emerging as promising alternatives to antibiotics, there are not yet any probiotics approved as a treatment in dental practice and substantial clinical research must still be undertaken. Among the huge variety of probiotics, different probiotic strains may have different mechanisms and, consequently, may cause different effects on the different types of oral infections (e.g., periodontitis, dental caries, and peri-implantitis), which are associated with a specific microbiome. Therefore, the clinical indication of specific probiotic strains for specific oral pathologies is still to be determined [137]. Since probiotics are living microorganisms and not drugs, standard pharmacokinetic assays do not apply, and many specific features and effects have to be ascertained in vivo and through well-designed clinical trials. Specifically, the most effective probiotic combinations, the probiotic ability to colonize the oral cavity and/or integrate certain oral biofilms, and its survival in the oral cavity, as well as the most appropriate probiotic vehicle, the exact dose, the frequency of administration, and the duration of treatment, need to be evaluated [135,136].

## 9. Conclusions

Antibiotic resistance is a ubiquitous issue in all health medical fields, including Dental Medicine. This public health problem has been tackled through antibiotic stewardship programs. Alternative strategies to antibiotic treatment in dental practice are being largely investigated, such as the use of antimicrobial photodynamic therapy (APDT), cold atmospheric plasma (CAP), probiotics, natural products, inhibitors of virulence factors, and antimicrobial peptides. These alternatives are still in the pre-clinical stages, thus requiring clinical approval and efficacy confirmation.

## Figures and Tables

**Figure 1 ijerph-19-16448-f001:**
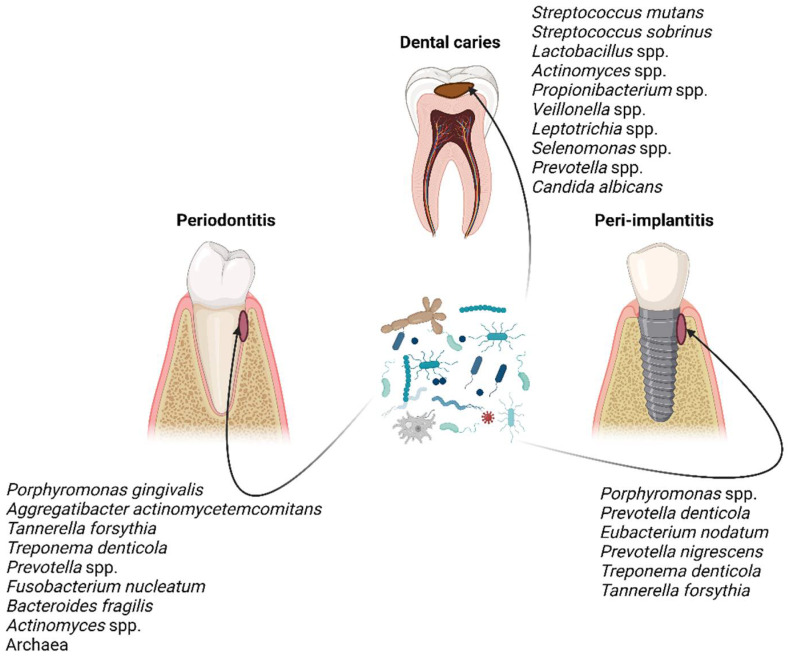
Characteristic taxa associated with dental caries, periodontitis, and peri-implantitis [30,31,32,33,34,35]. Figure created with BioRender.com.

**Figure 2 ijerph-19-16448-f002:**
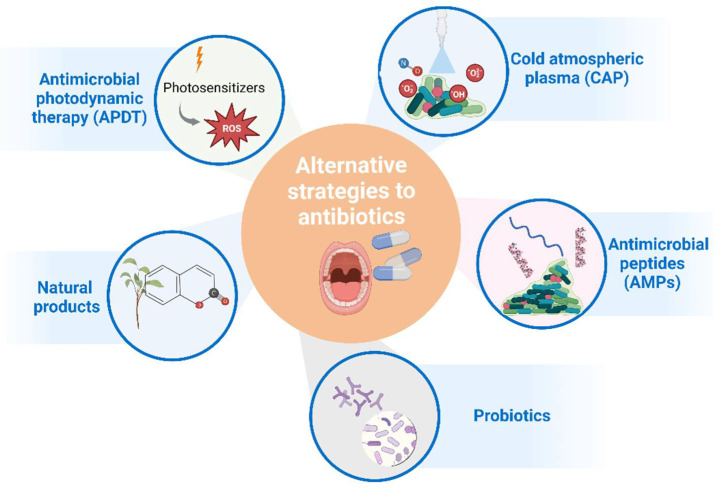
Main alternative strategies to antibiotic treatment that have been considered for clinical application in oral infections. Figure created with BioRender.com.

**Table 1 ijerph-19-16448-t001:** Antibiotic stewardship interventions in dental practice.

• Ponder patient conditions and look for a clear diagnosis before prescribing antibiotics; discuss with peers and other specialists if needed
• Follow updated and standardized guidelines
• Receive feedback on previous acts of antibiotic prescribing
• Warrant ongoing education and appropriate training
• Educate the dental patient and establish good communication to ensure the patient will follow the correct instructions when taking antibiotics
• Audit how appropriately antimicrobials are prescribed

These points have been proposed by recent studies [56,68,70].

## Data Availability

Not applicable.

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
