# Peer review of "Managing Oral Health in the Context of Antimicrobial Resistance"

_ijerph, 2022, doi:10.3390/ijerph192416448_

Round 1
Reviewer 1 Report
Dear Authors
I would like to congratulate you for the elaboration of this review, I found it a wonderful update and now I understand the extended amount of very updated bibliographical references that you incorporate.
I would just like you to indicate in bold the words or concepts that are going to be defined, such as microbiona, dental plaque, resistome, it is important for the reader to look for them and to be able to reference the article correctly.
I loved the content as I have already commented, and perhaps I would introduce a table with the most common antibiotic indications (types and doses) in dentistry that are published as they say in articles 57, 61-66. This would also serve as a reference for this article in future publications.
My sincere congratulations
Author Response
Reviewer #1:
R1: “I would like to congratulate you for the elaboration of this review, I found it a wonderful update and now I understand the extended amount of very updated bibliographical references that you incorporate.”
Authors: We thank the reviewer for his positive feedback on our review. We appreciate the kind words.
R1: “I would just like you to indicate in bold the words or concepts that are going to be defined, such as microbiona, dental plaque, resistome, it is important for the reader to look for them and to be able to reference the article correctly.”
Authors: As suggested, we have highlighted in bold relevant terms such as “oral microbiome”, “oral resistome”, “dental caries”, “periodontitis” and “peri-implantitis” in the parts of the manuscript we found more pertinent so that readers are able to understand the definition of such terms.
R1: “I loved the content as I have already commented, and perhaps I would introduce a table with the most common antibiotic indications (types and doses) in dentistry that are published as they say in articles 57, 61-66. This would also serve as a reference for this article in future publications.”
Authors: We appreciate the suggestion, but as we say in the text “guidelines for prudent antibiotic use can vary among countries, recommending different approaches based on the antibiotic resistance patterns known nationwide”. “Also, therapeutic and prophylactic indications for dental antibiotics are not consensual and differ internationally”.
Therefore, we cannot construct a table as the reviewer suggests, because there are no definite antibiotic indications in dentistry and commercial presentations (doses available) vary from country to country; such a table could be misleading.
Reviewer 2 Report
I want to congratulate the authors on an excellent comprehensive review. However, it isn't easy to read due to the writing structure. We need more tables to understand better. In my opinion, please add the tables under each chapter to summarize describing papers. The table should include 1) the name of the first author et al., year and no of reference 2) the type of the study - original/clinical study etc. 3) the aim; 4) the main findings 2. Even if it is a narrative review, the methods section should be added - criteria for searching the articles (database, years, type of the studies, key words). 3. please add strength and limitations sectionAuthor Response
Reviewer #2
R2: “I want to congratulate the authors on an excellent comprehensive review.”
Authors: We thank the reviewer for his positive feedback on our review.
R2: “However, it isn't easy to read due to the writing structure. We need more tables to understand better. In my opinion, please add the tables under each chapter to summarize describing papers. The table should include 1) the name of the first author et al., year and no of reference 2) the type of the study - original/clinical study etc. 3) the aim; 4) the main findings”
Authors: In this review, we have two figures and one table. We understand that tables and figures are useful and attractive to the readers. Nonetheless, since this is a narrative review and not a systematic review, there are no describing papers that we should particularize and summarize in tables as suggested by the reviewer. There are no reference papers being compared. Therefore, we cannot comply with the reviewer’s suggestion, because doing so would be counterintuitive.
R2: “Even if it is a narrative review, the methods section should be added - criteria for searching the articles (database, years, type of the studies, keywords).”
Authors: We appreciate the reviewer’s suggestion. We have inserted that section in the revised manuscript (lines 69-75).
R2: “3. please add strength and limitations section”
Authors: Since this is a narrative review and not a systematic review, we are not in the position to indicate strengths and limitations.
Reviewer 3 Report
The authors have provided an updated overview of how dentists are adapting to better manage the treatment of oral infections given the problem of AMR
The authors list the bacteria present in dental caries, periodontitis, and peri-implantitis. But it is necessary to provide a brief on how they differentiate.
doi: 10.1055/s-0040-1715779
doi: 10.2147/TCRM.S139795
It would be useful if can add recent techniques on biofilm prevention.
doi: 10.3390/nano10061230
https://doi.org/10.1038/s41598-021-92340-7
If possible add a table on antimicrobial resistance and management with time.
Author Response
Reviewer #3
“The authors have provided an updated overview of how dentists are adapting to better manage the treatment of oral infections given the problem of AMR”.
R3: “The authors list the bacteria present in dental caries, periodontitis, and peri-implantitis. But it is necessary to provide a brief on how they differentiate.”
doi: 10.1055/s-0040-1715779
doi: 10.2147/TCRM.S139795
Authors: In Figure 1, we listed the most common taxa associated with dental caries, periodontitis, and peri-implantitis, according to several studies performed over the years, which are cited (references 30-35). Therefore, there is evidence that the microbial communities are distinct in dental caries, periodontitis, and peri-implantitis in terms of composition and/or abundance. Nonetheless, as we also pinpoint in the text, the taxonomic composition implicated in each mentioned disease is not definite and is being constantly updated as knowledge and technologies are advancing. In the revised manuscript, we added a new sentence in order to better clarify this topic (lines 122-124). We appreciate the papers suggested by the reviewer that are related to the microbiome of peri-implantitis. However, we could not manage to insert them, since they would not add any additional support to what has been stated in the references cited, such as 32 and 35. Moreover, in the following section “Oral biofilms and biofilm-related oral diseases”, we better differentiate those three diseases and highlight the relevant bacteria implicated in each one.
R3: “It would be useful if can add recent techniques on biofilm prevention.
doi: 10.3390/nano10061230
https://doi.org/10.1038/s41598-021-92340-7”
Authors: Accordingly, we have added the suggested information and references (lines 404 and 406). Thus, two new references were added, now numbered 106 and 107.
R3: If possible add a table on antimicrobial resistance and management with time.
Authors: Antimicrobial resistance is an evolving and complex process that happens naturally, however, the misuse and overuse of antibiotics have accelerated the process. Strategies to deal with and manage antibiotic resistance in the dental practice have been addressed throughout some sections of the manuscript, namely “Antibiotic stewardship in dentistry” and “Current alternatives to antibiotics to prevent and treat oral infections”. We struggled in understanding the suggestion of the reviewer to add a table on antimicrobial resistance and management with time. It would be hard to summarize in a table the management of antimicrobial resistance with time since that is a complex, broad, and in constant adaptation topic.
Reviewer 4 Report
I congratulate the authors on an interesting review paper. The topic of the manuscript is above all current and important. I am sorry that this manuscript was not made as a systematic review with meta analysis, because I think that would be better. But all in all, this is an interesting narrative review.
Author Response
Reviewer #4
R4: “I congratulate the authors on an interesting review paper. The topic of the manuscript is above all current and important. I am sorry that this manuscript was not made as a systematic review with meta analysis, because I think that would be better. But all in all, this is an interesting narrative review.”
Authors: We thank the reviewer for acknowledging the relevance of the topic. Our purpose was a narrative review and not a systematic review.